# BMP7 functions predominantly as a heterodimer with BMP2 or BMP4 during mammalian embryogenesis

Hyung-Seok Kim[1], Judith Neugebauer[1], Autumn McKnite[1], Anup Tilak[2], Jan L Christian[1]*

[1]Department of Neurobiology and Anatomy and Internal Medicine, Division of Hematology and Hematologic Malignancies, School of Medicine, University of Utah, Salt Lake City, United States; [2]Department of Cell and Developmental Biology, School of Medicine, Oregon Health and Sciences University, Portland, United States

**Abstract** BMP7/BMP2 or BMP7/BMP4 heterodimers are more active than homodimers in vitro, but it is not known whether these heterodimers signal in vivo. To test this, we generated knock in mice carrying a mutation ($Bmp7^{R-GFlag}$) that prevents proteolytic activation of the dimerized BMP7 precursor protein. This mutation eliminates the function of BMP7 homodimers and all other BMPs that normally heterodimerize with BMP7. While $Bmp7$ null homozygotes are live born, $Bmp7^{R-GFlag}$ homozygotes are embryonic lethal and have broadly reduced BMP activity. Furthermore, compound heterozygotes carrying the $Bmp7^{R-G}$ allele together with a null allele of $Bmp2$ or $Bmp4$ die during embryogenesis with defects in ventral body wall closure and/or the heart. Co-immunoprecipitation assays confirm that endogenous BMP4/7 heterodimers exist. Thus, BMP7 functions predominantly as a heterodimer with BMP2 or BMP4 during mammalian development, which may explain why mutations in either $Bmp4$ or $Bmp7$ lead to a similar spectrum of congenital defects in humans.

DOI: https://doi.org/10.7554/eLife.48872.001

*For correspondence:
jan.christian@neuro.utah.edu

Competing interests: The authors declare that no competing interests exist.

## Introduction

Bone morphogenetic proteins (BMPs) are secreted molecules that were initially discovered as bone inducing factors and were subsequently shown to play numerous critical roles during embryogenesis (*Bragdon et al., 2011*). Recombinant BMPs are used clinically to treat bone loss caused by trauma or disease, but their usefulness as osteoinductive agents is limited by a short half-life when implanted in vivo (*Khan et al., 2012*). Understanding how BMP dosage is regulated in vivo is important to prevent congenital birth defects, and to aid in the development of more effective therapeutics to promote bone healing.

BMPs are grouped into subfamilies based on sequence similarity, and can signal as either homodimers or as heterodimers. The class I BMPs, BMP2 and BMP4, can heterodimerize with class II BMPs, consisting of BMPs 5–8 (*Guo and Wu, 2012*). Heterodimers composed of class I and class II BMPs show a higher specific activity than do homodimers. For example, homodimers of BMP2, −4, or-7 can all induce bone formation, but BMP2/7 or BMP4/7 heterodimers are significantly more potent than any homodimer in osteogenic differentiation assays (*Aono et al., 1995*; *Kaito et al., 2018*). Likewise, BMP2/6 heterodimers show enhanced ability to activate downstream signaling in embryonic stem cells (*Valera et al., 2010*). BMP2/7 and BMP4/7 heterodimers also show enhanced ability to induce ventral fate in *Xenopus* and zebrafish (*Nishimatsu and Thomsen, 1998*; *Schmid et al., 2000*).

While it is widely accepted that recombinant class I/II BMP heterodimers have higher specific activity than homodimers, whether endogenous BMPs function primarily as homodimers or heterodimers in vivo remains controversial. Mutations in either *Bmp2b* or *Bmp7* lead to a complete loss of signaling in zebrafish embryos and this can be rescued by recombinant heterodimers, but not by either homodimer (*Little and Mullins, 2009*). In addition, an ectopically expressed epitope-tagged form of BMP2b pulls down endogenous BMP7 and vice versa (*Little and Mullins, 2009*). Thus, BMP2/7 heterodimers are essential to establish the dorsoventral axis in fish. In *Drosophila,* DPP (the fly homolog of BMP2/4) is not properly localized to the embryonic midline in the absence of SCREW (a BMP7 homolog) and this is proposed to be due to preferential transport of DPP/SCREW heterodimers (*Shimmi et al., 2005*). However, expression of DPP and SCREW homodimers in distinct regions of the embryo can activate BMP signaling at levels equivalent to the heterodimer (*Nguyen et al., 1998*; *Wang and Ferguson, 2005*), suggesting that homodimers are sufficient for development.

Evidence that class I/II BMP heterodimers exist or are required for mammalian development is lacking. *Bmp4* or *Bmp2* null homozygotes die during early development with defects in multiple tissues that correlate well with their respective expression domains (*Winnier et al., 1995*; *Zhang and Bradley, 1996*). Among Class II BMPs, *Bmp8* is restricted to the developing testes and placenta while *Bmp5*, *Bmp6* and *Bmp7* are broadly expressed throughout embryogenesis (*Zhao, 2003*). Mice homozygous for null mutations in any single Class II *Bmp* gene survive embryogenesis (*King et al., 1994*; *Dudley et al., 1995*; *Luo et al., 1995*; *Solloway et al., 1998*). By contrast, *Bmp5;Bmp7* or *Bmp6;Bmp7* double mutants are embryonic lethal (*Solloway and Robertson, 1999*; *Kim et al., 2001*), demonstrating functional redundancy. *Bmp2/7* and *Bmp4/7* double heterozygotes present with no abnormalities or minor skeletal abnormalities (*Katagiri et al., 1998*), raising the possibility that heterodimers are not required for early mammalian development.

The choice of whether a given BMP will form a homodimer or a heterodimer is made within the biosynthetic pathway. All BMPs are generated as inactive precursor proteins that dimerize and fold within the endoplasmic reticulum (*Bragdon et al., 2011*). The precursor protein is then cleaved by members of the proprotein convertase (PC) family to generate the active, disulfide bonded ligand along with two prodomain fragments. We have shown that BMP4 and BMP7 preferentially form heterodimers rather than either homodimer when ectopically coexpressed in *Xenopus* embryos (*Neugebauer et al., 2015*). *Bmp4 and Bmp7* show overlapping patterns of expression in many tissues (*Danesh et al., 2009*), suggesting that they may form heterodimers in some contexts. In humans, heterozygous mutations in either *Bmp4* or *Bmp7* are associated with a similar spectrum of ocular, brain and palate abnormalities (*Bakrania et al., 2008*; *Suzuki et al., 2009*; *Wyatt et al., 2010*; *Reis et al., 2011*), consistent with the possibility that mutations in either gene lead to reduced BMP4/7 heterodimer activity.

BMPs carrying mutations in the PC cleavage motif form inactive dimers with wild type proteins. For example, cleavage mutant forms of BMP7 dominantly interfere with both BMP7 and BMP4 signaling when overexpressed in *Xenopus* (*Hawley et al., 1995*; *Nishimatsu and Thomsen, 1998*). These studies demonstrate that BMPs can heterodimerize when overexpressed, but do not address whether endogenous BMPs heterodimerize. A mouse carrying a point mutation in the cleavage site of the type II BMP, BMP5, has more severe skeletal abnormalities than *Bmp5* homozygous null mutants (*Ho et al., 2008*), consistent with the possibility that it interferes with Class I heterodimeric partners, but this has not been explored.

In the current studies, we used genetic and biochemical analysis to test the hypothesis that endogenous heterodimers containing BMP7 exist in vivo. We show that endogenous heterodimers do form in vivo, and are the predominant functional ligand in many if not all tissues of developing mouse embryos. These findings have relevance to understanding the impact of mutations in *Bmp4* or *Bmp7* in humans.

## Results

### *Bmp7*<sup>*R-GFlag*</sup> homozygotes show earlier lethality and more severe phenotypic defects than *Bmp7* null homozygotes

The phenotypes observed in mice mutant for class I and/or class II *Bmps* can be explained if BMPs function either as homodimers, or as heterodimers (illustrated in *Figure 1A*). In the hypothetical homodimer model (top), all BMP activity within a given cell type is generated by homodimers of class I (BMP2 or 4) and class II BMPs (BMP5, 6 or 7) that are broadly expressed in development. In the competing heterodimer model (bottom), all of the available class I molecules are covalently dimerized with a class II molecule, and these heterodimers generate the majority of BMP activity within a specific cell type. In addition, a small pool of 'excess' class II BMPs is hypothesized to form homodimers under wild type conditions, but is available to form heterodimers to compensate for loss of any single class II BMP. This hypothetical model is consistent with our finding that BMPs preferentially form heterodimers in some contexts (*Neugebauer et al., 2015*), and with the genetic redundancy observed among class II BMPs. It is not meant to imply that this happens in all cell types.

To ask whether heterodimers containing endogenous BMP7 are required for normal development, we generated a *Bmp7* cleavage mutant mouse (*Bmp7*<sup>*R-GFlag*</sup>) (*Figure 1—figure supplement 1*). These mice have a point mutation that changes the amino acid sequence of the PC cleavage motif from -RISR- to -RISG-, as well as sequence encoding a Flag epitope tag knocked in to the *Bmp7* allele (*Figure 1B*). *Bmp7*<sup>*R-GFlag*</sup> mice express endogenous levels of a non-cleavable, inactive BMP7 precursor protein. A control mouse that carries the Flag-epitope tag at the wild type *Bmp7* locus (*Bmp7*<sup>*Flag*</sup>) was generated in parallel.

If BMP7 signals exclusively as a homodimer in all cells, then *Bmp7*<sup>*R-GFlag*</sup> homozygotes should show the same reduction in BMP activity as *Bmp7* null mutants (illustrated in *Figure 1B–C*; upper diagrams), and would be predicted to die shortly after birth due to kidney defects (*Dudley et al., 1995*; *Luo et al., 1995*). By contrast, if class I/II heterodimers are the primary functional ligand, then the BMP7R-GFlag precursor protein would form non-functional covalent heterodimers with all endogenous class I BMPs with which it normally interacts (*Figure 1B*, lower diagram). In this case, the hypothetical 'surplus pool' of class II BMPs that can buffer the heterodimer pool in *Bmp7* null mutants (*Figure 1C*, lower left) will be unable to compensate in *Bmp7*<sup>*R-GFlag*</sup> mutants (*Figure 1C*, lower right). This would lead to a greater reduction in the heterodimer pool, lower total BMP activity and more severe phenotypic defects in *Bmp7*<sup>*R-GFlag*</sup> than in then *Bmp7* null mutants in any tissues or cell types where heterodimers dominate.

*Bmp7*<sup>*R-GFlag/+*</sup>, *Bmp7*<sup>*Flag/+*</sup> or *Bmp7*<sup>*-/+*</sup> mice were intercrossed to determine viability. *Bmp7*<sup>*Flag/Flag*</sup> embryos were recovered at the predicted Mendelian frequency throughout development and were adult viable with no apparent defects (*Table 1*). *Bmp7* null homozygotes were recovered at the predicted Mendelian ratio between embryonic day (E)9.5 and E18.5 (*Table 2*) but died shortly after birth. By contrast, *Bmp7*<sup>*R-GFlag/R-GFlag*</sup> mice were present at the predicted Mendelian frequency through E11.5 but were not recovered after this stage (*Table 3*).

*Bmp7*<sup>*Flag/Flag*</sup> (*Figure 1D,D', G,G', J,J'*) and *Bmp7*<sup>*-/-*</sup> (*Figure 1F,F', I,I', L,L'*) embryos were indistinguishable from wild type littermates at E9.5–11.5, with the exception of slightly smaller eyes in 25% of the *Bmp7*<sup>*-/-*</sup> embryos examined at E11.5 (*n* = 7; *Figure 1L,L'*). *Bmp7*<sup>*R-GFlag/ R-GFlag*</sup> embryos appeared grossly normal but were slightly smaller than age matched (by somite number) wild type littermates at E9.5 (*n* = 45; *Figure 1E,E'*). *Bmp7*<sup>*R-GFlag*</sup> homozygotes were smaller, and had smaller limb buds than littermates at E10.5 (*n* = 13; *Figure 1H,H'*, *Figure 1—figure supplement 2*) and were resorbing at E11.5 (*n* = 12; *Figure 1K,K'*). All *Bmp7*<sup>*R-GFlag*</sup> homozygotes showed multiple abnormalities at E10.5 (*Figure 1H,H'*) including smaller and less distinct forebrain (fb), midbrain (mb) and hindbrain (hb), pericardial edema (arrows), smaller limb buds (flb) and no eye (e). Thus, expression of wild type levels of an uncleavable BMP7 precursor protein leads to earlier lethality and more severe phenotypic defects than does complete absence of BMP7 protein, suggesting that endogenous BMP7-containing heterodimers perform essential functions during early embryogenesis.

*Bmp7*<sup>*R-GFlag*</sup> heterozygotes were adult viable (*Supplementary file 1*), but 23% showed runting, microphthalmia and/or anophthalmia as early as E14.5 (*n* = 13, *Figure 1—figure supplement 3A–B*) that persisted into adulthood (*Figure 1—figure supplement 3E–F*). These defects were never observed in *Bmp7* null heterozygotes (*n* = 8; *Figure 1—figure supplement 3C*) although one or

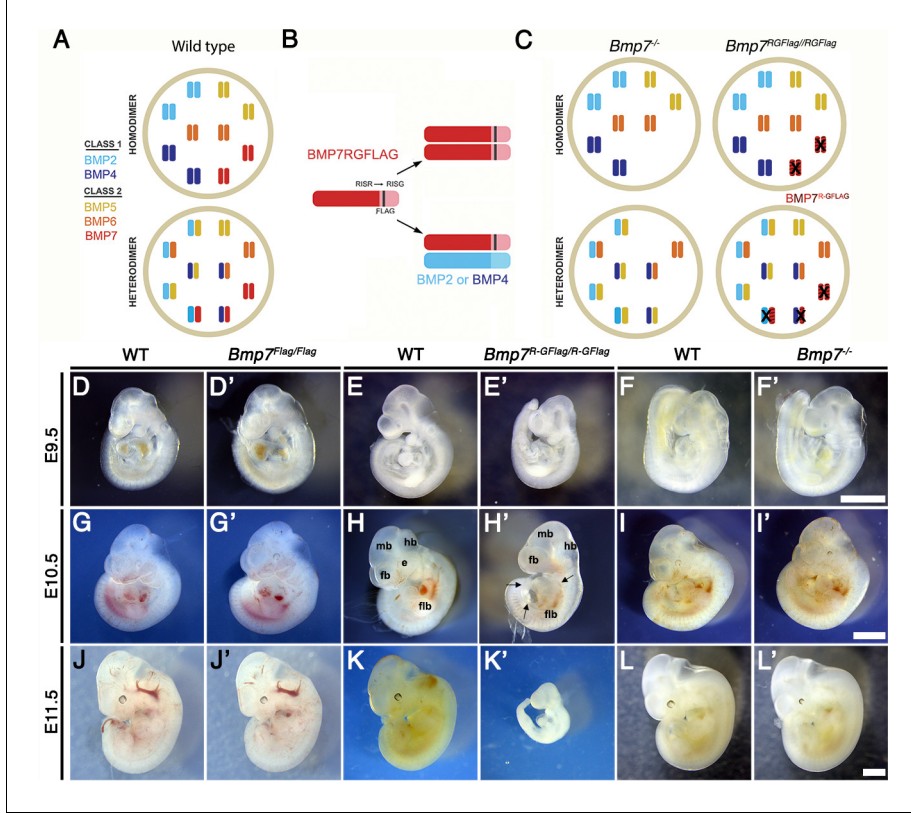

**Figure 1.** *Bmp7^{R-GFlag}* homozygotes show earlier lethality and more severe phenotypic defects than *Bmp7* null homozygotes. (**A**) Illustration of two hypothetical models in which class I and II BMPs function predominantly as homodimers (top) or as heterodimers (bottom) within select cell types. In the heterodimer model, class I/II heterodimers form preferentially. In addition, it is hypothesized that there is an excess of class II BMPs that form homodimers in the wild type condition but are available to form heterodimers to compensate for loss of any single class II BMP. (**B**) Illustration of BMP7R-GFlag precursor protein forming homodimers (top) or heterodimers with BMP2 or BMP4 (bottom). Prodomain: dark shading, mature domain: light shading, black bar: FLAG epitope. (**C**) Illustration showing predicted loss of BMP activity in *Bmp7^{R-GFlag}* or *Bmp7* null homozygotes if BMPs function predominantly as homodimers (top) or as heterodimers (bottom). In cells in which only homodimers form, there is predicted to be an equivalent reduction in BMP activity in *Bmp7^{-/-}* and in *Bmp7^{R-GFlag/Flag}* mice because only BMP7 homodimers are absent or inactive (black X), respectively. In cells or tissues in which class I/II heterodimers are the primary functional ligand, excess class II molecules that normally form homodimers will fill in to maintain the heterodimer pool in *Bmp7* null mutants (lower left), but cannot do so in *Bmp7^{R-GFlag}* mutants because the BMP7R-GFlag precursor protein forms non-functional covalent heterodimers with endogenous class I BMPs (lower right). This would lead to a greater reduction in the heterodimer pool, lower total BMP activity and more severe phenotypic defects in *Bmp7^{R-GFlag}* than in then *Bmp7* null mutants in any tissues or cell types where heterodimers predominate. (**D–L'**) Photograph of E9.5–11.5 (age indicated to left of each row) wild type (**D–L**) or mutant (D'-L'; genotype listed at top of each column) littermates. Scale bars in Panel F', I' and L' correspond to 1 mm and apply across each row. A minimum of eight embryos of each genotype were analyzed at each stage. Arrows in H' indicate pericardial edema. fb; forebrain, mb; midbrain, hb; hindbrain, flb; forelimb bud, e; eye.

DOI: https://doi.org/10.7554/eLife.48872.002

The following figure supplements are available for figure 1:

**Figure supplement 1.** Generation of *Bmp7^{R-GFlag}* mice.
DOI: https://doi.org/10.7554/eLife.48872.003

**Figure supplement 2.** Limb bud size is reduced in *Bmp7^{R-GFlag}* homozygotes.
DOI: https://doi.org/10.7554/eLife.48872.004

**Figure supplement 3.** *Bmp7^{R-GFlag}* heterozygotes show skeletal and eye defects that are absent in *Bmp7* null heterozygotes.
DOI: https://doi.org/10.7554/eLife.48872.005

**Table 1.** Progeny from *Bmp7^Flag/+* intercrosses

| Age | Bmp7+/+ | Bmp7Flag/+ | Bmp7Flag/Flag | Total |
|-----|---------|------------|---------------|-------|
| P28 | 8 (20%) | 24 (60%) | 8 (20%) | 40 |
| E9.5-14.5 | 23 (31%) | 34 (46%) | 17 (23%) | 74 |

Data are presented as number (percent).
DOI: https://doi.org/10.7554/eLife.48872.006

both eyes were absent in late gestation *Bmp7* null homozygotes (*n* = 5; ***Figure 1—figure supplement 3D***) as previously reported (***Dudley et al., 1995***; ***Luo et al., 1995***). Skeletal analysis revealed that the fibula was shortened and failed to articulate with the knee in 13% of *Bmp7^R-GFlag^* heterozygotes that were analyzed (*n* = 14; ***Figure 1—figure supplement 3G,H***). This defect has been observed in mice in which both *Bmp2* and *Bmp7* are conditionally deleted from the limbs, but not in mice lacking any single BMP family member (***Bandyopadhyay et al., 2006***). These findings support a model in which the BMP7R-GFlag precursor protein dominantly sequesters endogenous class I BMPs in non-functional dimers.

## *Bmp7^R-GFlag^* homozygotes show defects in heart development that are absent in *Bmp7* null homozygotes

*Bmp2, 4, and 7* are expressed in overlapping domains of the developing heart (***Dudley and Robertson, 1997***; ***Danesh et al., 2009***), raising the possibility that heart defects cause embryonic lethality in *Bmp7^R-GFlag/R-GFlag^* mutants. At E10.5, hearts dissected from *Bmp7^Flag^* (*n* = 5) and *Bmp7* null homozygotes (*n* = 6) were indistinguishable from wild type littermates with morphologically distinguishable atria, ventricles, and outflow tract (OFT) (***Figure 2A–A', C–C'***). By contrast, all hearts of *Bmp7^R-GFlag/R-GFlag^* embryos (*n* = 6) appeared to have thinner walls, had a common atrium, a smaller right ventricle and a small, malformed OFT relative to wild type littermates (***Figure 2B–B'***). In addition, although hearts of *Bmp7^R-GFlag/R-GFlag^* mutants were morphologically normal at E9.5, expression of the direct BMP target gene, *Nkx2.5* (***Lien et al., 2002***), was severely reduced in all hearts of *Bmp7^R-GFlag/R-GFlag^* mutants relative to littermates at E9.5 (*n* = 10) and E10.5 (*n* = 6) (***Figure 2E–E', H–H', K–K', N–N'***). No differences were detected in expression of *Nkx2.5* in *Bmp7^Flag^* (***Figure 2D–D', G–G', J–J', M–M'***) or in *Bmp7* null homozygotes (***Figure 2F–F', I–I', L–L', O–O'***) relative to wild type littermates.

## *Bmp7^R-GFlag^* mutants, but not *Bmp7* null mutants, show reduced BMP activity in multiple tissues

To test whether BMP activity is more severely reduced in *Bmp7^R-GFlag^* mutants than it is in *Bmp7* null mutants, we analyzed BMP activity in *BRE:LacZ* transgenic embryos at E9.5, before gross morphological abnormalities are detected in *Bmp7^R-GFlag^* homozygotes. This transgene contains a BMP-responsive element coupled to LacZ, which serves as an in vivo reporter of BMP signaling downstream of all endogenous BMP ligands (***Monteiro et al., 2004***). X-GAL staining for ß-galactosidase activity in *Bmp7^+/+^;BRE:LacZ* embryos revealed strong endogenous BMP activity in the brain, eye, branchial arches (BA), heart, and ventroposterior mesoderm (VPM) (***Figure 3A,C***). No differences in BMP activity were detected in any of these tissues in *Bmp7^-/-^* embryos (***Figure 3A,B***). By contrast, as shown in ***Figure 3C and D***, *Bmp7^R-GFlag/R-GFlag^* embryos exhibited a reproducible reduction in BMP activity in the brain, ventroposterior mesoderm, and heart. The reduction in staining in the heart of *Bmp7^R-GFlag/R-GFlag^* embryos was most pronounced in the right ventricle (outlined in white in the

**Table 2.** Progeny from *Bmp7^-/+* intercrosses

| Age | Bmp7+/+ | Bmp7-/+ | Bmp7-/- | Total |
|-----|---------|---------|---------|-------|
| E13.5-18.5 | 12 (30%) | 18 (45%) | 10 (25%) | 40 |
| E9-12.5 | 21 (32%) | 32 (48%) | 13 (20%) | 66 |

Data are presented as number (percent).
DOI: https://doi.org/10.7554/eLife.48872.007

**Table 3.** Progeny from *Bmp7^R-GFlag/+* intercrosses

| Age | Bmp7+/+ | Bmp7R-GFlag/+ | Bmp7R-GFlag/R-GFlag | Total |
|---|---|---|---|---|
| E18.5-P0** | 15 (52%) | 14 (41%) | 0 (0%) | 29 |
| E12.5-E14.5* | 9 (47%) | 11 (53%) | 0 (0%) | 20 |
| E11.5 | 12 (29%) | 18(42%) | 12 (29%) | 42 |
| E10.5 | 6 (17%) | 16 (46%) | 13 (37%) | 35 |
| E9.5 | 28 (18%) | 83 (53%) | 45 (29%) | 156 |

Data are presented as number (percent). Asterisks indicate that the observed frequency is significantly different than the expected frequency by Chi-square analysis (**P<0.01).
DOI: https://doi.org/10.7554/eLife.48872.008

inset) and the OFT (outlined in magenta in inset). In addition, staining was completely absent in the eye (*Figure 3D*). Thus, mice expressing endogenous levels of an uncleavable form of the BMP7 precursor protein show widespread loss of BMP activity that is not observed in mice lacking BMP7 protein.

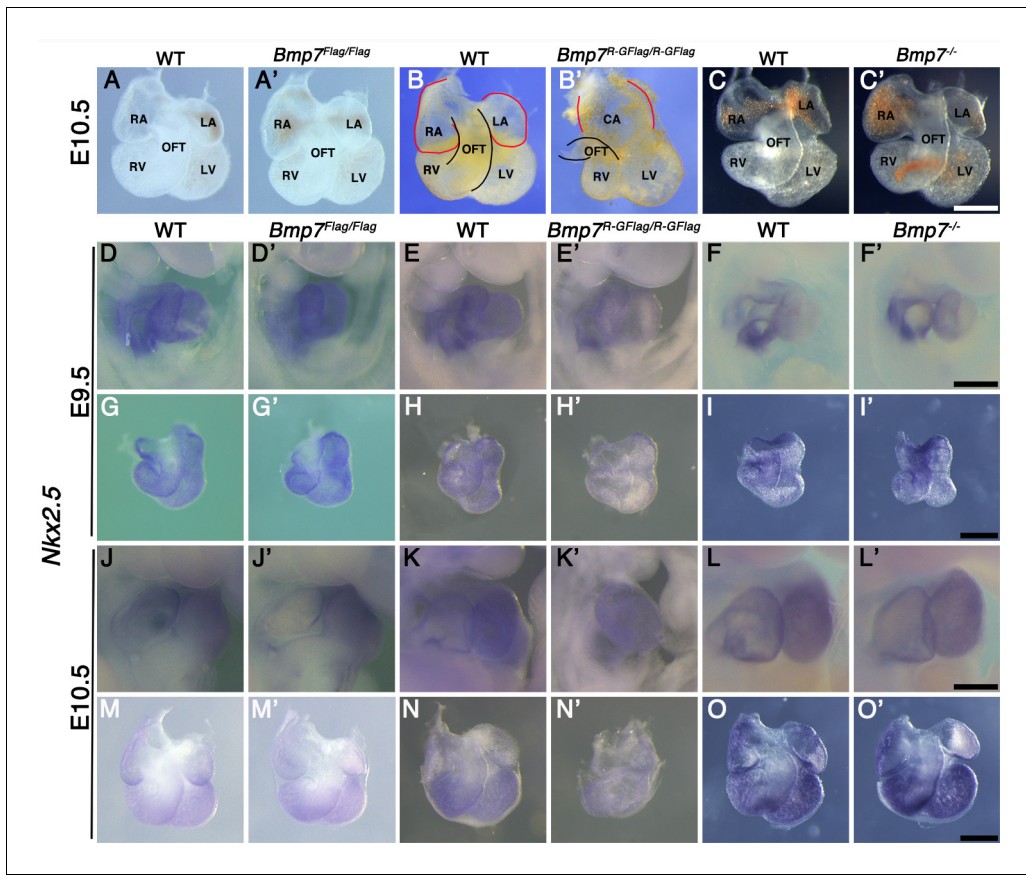

**Figure 2.** *Bmp7^R-GFlag* homozygotes show defects in heart development that are absent in *Bmp7* null homozygotes. (**A–C'**) Photographs of hearts dissected from E10.5 mice carrying targeted alleles of *Bmp7* (**A'–C'**) and wild type littermates (**A–C**). Genotypes are indicated above each panel. The OFT is outlined in black, and the atrium in red in B and B'. (**D–O'**) Expression of *Nkx2.5* was analyzed by whole mount in situ hybridization in mice carrying targeted alleles of *Bmp7* (**D'–O'**) and wild type littermates (**D–O**) at E9.5 or E10.5 as indicated to the left of each row. Genotypes are indicated above each panel. Close up photographs of hearts in intact embryos (**D–F'**, **J–L'**) or dissected free of embryos (**G–I'**, **M–O'**) are shown. RA; right atrium, LA; left atrium, CA; common atrium, RV; right ventricle, LV; left ventricle, OFT; outflow tract. Scale bars in all panels correspond to 500 µM.
DOI: https://doi.org/10.7554/eLife.48872.009

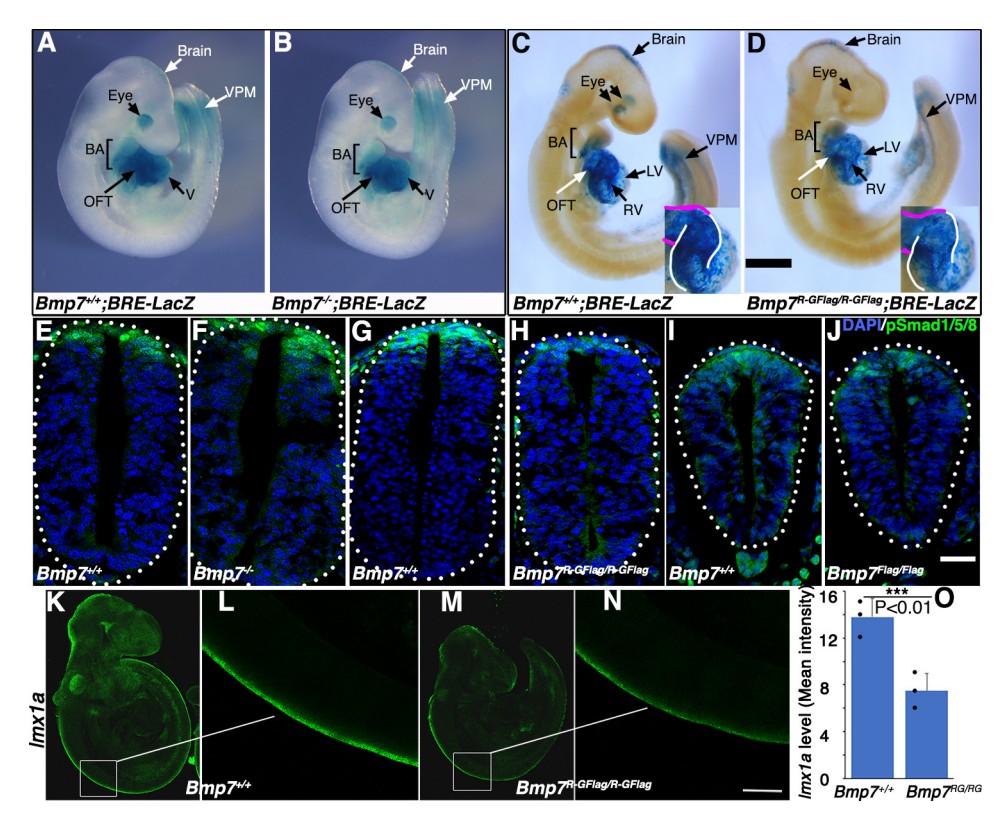

**Figure 3.** *Bmp7^{R-GFlag}* mutants, but not *Bmp7* null mutants, show reduced BMP activity in multiple tissues. **(A–D)** E9.5 *Bmp7^{-/-}* **(B)**, *Bmp7^{R-GFlag/R-GFlag}* **(D)** and wild type littermates **(A, C)** carrying a BRE-LacZ transgene were stained for ß-galactosidase activity to detect endogenous BMP pathway activation. Embryos from a single litter were stained for an identical period of time under identical conditions. A minimum of three embryos of each genotype were examined and results shown were reproduced in all. VPM; ventral posterior mesoderm, BA; branchial arches, OFT; outflow tract, RV; right ventricle, LV; left ventricle V; ventricles. Insets in C, D show an enlarged view of the heart with the OFT and RV outlined in white and magenta, respectively. In C, left and right eyes are visible in the cleared embryo. Scale bars correspond to 1 mm. **(E–H)** Transverse sections through the spinal cord of E9.5 *Bmp7^{-/-}* **(F)**, *Bmp7^{R-GFlag/R-GFlag}* **(H)**, *Bmp7^{Flag/Flag}* **(J)**, and wild type littermates **(E, G, I)** were immunostained with antibodies specific for pSmad1/5/8 and nuclei were stained with DAPI. Dorsal is up. Scale bars correspond to 20 μm. Three embryos of each genotype were analyzed and results were reproduced in all. **(K–N)** In situ HCR was used to analyze expression of *lmx1a* in three E9.5 *Bmp7^{R-GFlag/R-GFlag}* **(M–N)** and wild type littermates **(K–L)**. White boxes in K and M indicate the region of staining shown in L and N. Scale bar corresponds to 100 μm. **(O)** Levels of *lmxa1* transcripts (mean fluorescence ± s.d., data analyzed by two tailed *t*-test). Fluorescence intensity was quantified in comparable regions of the spinal cord in three embryos of each genotype. Dots on the graph denote values for individual embryos.
DOI: https://doi.org/10.7554/eLife.48872.010

---

*Bmp2*, *4* and *7* are co-expressed in the dorsal surface ectoderm overlying the spinal cord by E8 (*Solloway and Robertson, 1999*; *Danesh et al., 2009*). BMP signaling from the ectoderm is required for induction of the roof plate at E9, and BMPs and other factors secreted from the roof plate are subsequently required for specification, migration and axon guidance of dorsal interneurons (*Chizhikov and Millen, 2004*). To analyze BMP activity in this important signaling center, we immunostained sections of E9.5 embryos using an antibody specific for the active, phosphorylated form of BMP pathway-specific SMADs (pSmad1/5/8). Levels of pSmad1/5/8 were unchanged in the roof plate of *Bmp7^{-/-}* (*Figure 3E–F*) and *Bmp7^{Flag/Flag}* embryos relative to wild type littermates (*Figure 3I–J*), but were severely reduced in the roof plate of *Bmp7^{R-GFlag}* homozygotes (*Figure 3G–H*).

To further test whether BMP heterodimers are required for initial induction of the roof plate, we analyzed expression of *lmx1a* using in situ hybridization chain reaction (HCR). Expression of *lmx1a* is induced in the nascent roof plate downstream of BMP signaling from the epidermal ectoderm, and is the major mediator of BMP signaling in the dorsal neural tube (*Chizhikov and Millen, 2005*). Expression of *lmx1a* was reduced by 50% in all E9.5 *Bmp7^{R-GFlag/R-GFlag}* embryos that were examined (n = 3; *Figure 3K–O*). Thus, BMP7 containing heterodimers secreted from the epidermal ectoderm are major contributors to roof plate induction.

## Bmp7R-GFlag homodimers do not act outside of cells to block BMP activity

Our results support a model in which BMP7-containing class I/II heterodimers are the predominant functional ligand in early embryos. An assumption of this model is that the BMP7R-GFlag precursor protein forms covalent heterodimers with class I BMP precursor proteins inside of cells in which they are co-expressed (illustrated in *Figure 1B*), thus sequestering heterodimers in non-functional complexes that are unable to activate their receptors. An alternate possibility is that BMP7R-GFlag precursor forms uncleavable homodimers that are secreted and form non-functional complexes with BMP receptors on the cell surface, thereby blocking the ability of class I BMP homodimers to activate their cognate receptors. To test this possibility, we expressed BMP7R-GFlag or BMP4 in HEK293T cells, and collected BMP7R-GFlag precursor protein or mature BMP4 that was secreted into the culture medium. Non-transfected HEK293T cells were then exposed to equivalent amounts of mature BMP4 alone, BMP7R-GFlag precursor alone, or both together for one hour prior to analyzing levels of pSmad1/5/8 by immunoblot (illustrated in *Figure 4A*). Cells incubated with BMP7R-GFlag showed the same barely detectable level of immunoreactive pSmad1/5/8 as did control cells (*Figure 4B*, compare lane 1 and 3), indicating that the precursor protein lacks activity. Levels of pSmad1/5/8 were elevated to the same extent in cells incubated with BMP4 alone, or with BMP4 and BMP7R-GFlag together (compare lane 2 and 4). Thus, uncleaved BMP7R-GFlag precursor protein homodimers cannot act outside of the cell to block the ability of BMP4 to signal.

To further test whether BMP7R-GFlag precursor protein forms nonfunctional heterodimers with class I BMPs, we expressed BMP4 alone or together BMP7R-GFlag in HEK293T cells. Proteins in equivalent volumes of conditioned media were concentrated by trichloracetic acid precipitation, or were immunoprecipitated with antibodies specific for the Flag tag prior to blotting with antibodies specific for BMP4 (illustrated in *Figure 4C*). When BMP4 and BM7R-GFlag were co-expressed, relatively equivalent amounts of cleaved, mature BMP4 were detected in Flag immunoprecipitates and in input samples, suggesting that most BMP4 was heterodimerized with BMP7R-GFlag (*Figure 4D*, compare lanes 3 and 6). Steady state levels of mature BMP4 protein were much higher in the media of cells cotransfected with BMP4 and BMP7R-GFlag DNA than in an equivalent volume of media from cells transfected with the same amount of BMP4 DNA alone (compare lanes 2 and 3). The finding that BMP4/BMP7R-GFlag heterodimers accumulate to much higher levels than do BMP4 homodimers is consistent with an inability of the heterodimer to bind to, and induce internalization and degradation of the ligand/receptor complex, a process that is critical for normal embryogenesis (*Aoyama et al., 2012*). When untransfected HEK293T cells were exposed to media containing BMP4 homodimers, pSmad1 levels were increased 10-fold over basal levels (compare lanes 1 and 2). By contrast, exposure to the same volume of media from cells co-transfected with BMP4 and BMP7R-GFlag led to a 3.4-fold elevation of pSmad1 levels (compare lane 1 and 3), despite the fact that this media contains very high levels of cleaved BMP4 heterodimerized with BMP7R-GFlag. Taken together, these data demonstrate that the BMP4 precursor protein is cleaved and secreted when heterodimerized with the uncleavable BMP7R-GFlag precursor. However, this cleavage product is less able, or unable to bind and/or activate BMP receptors.

## Analysis of compound heterozygotes demonstrates that BMP2/7 and BMP4/7 heterodimers are functionally important during early development

To further test the idea that heterodimers of BMP7 together with BMP2 and/or BMP4 are essential for embryogenesis, and to ask which class I ligand(s) contribute to distinct developmental processes, we generated compound heterozygous mutants that carry one copy of the *Bmp7^{R-GFlag}* allele in

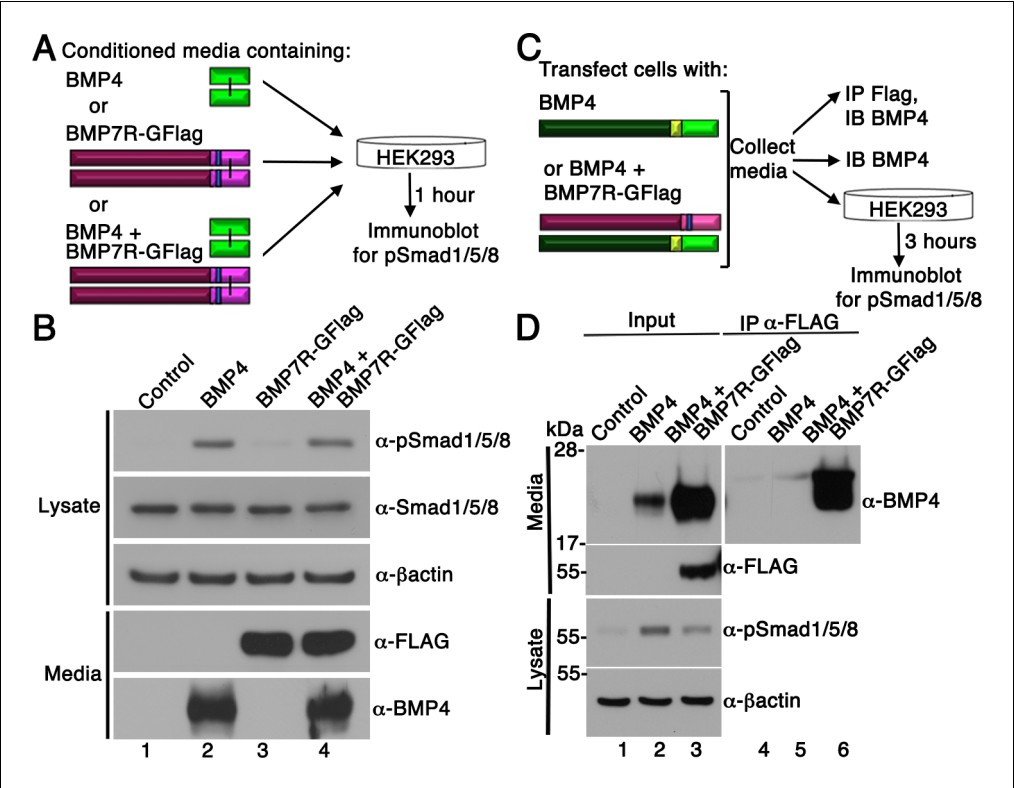

**Figure 4.** BMP7R-GFlag homodimers form inside of cells and cannot act outside of cells to block BMP signaling. (A) HEK293T cells were cultured for one hour in conditioned media containing equivalent amounts of BMP4 ligand alone, BMP7R-GFlag precursor protein alone or BMP4 ligand and BMP7R-GFlag precursor together as illustrated. (B) Proteins were separated by electrophoresis under reducing conditions and levels of pSmad1/5/8, total Smad1/5/8, BMP7R-GFlag and BMP4 (monomers) were analyzed by immunoblot. Blots were reprobed for ß-actin as a loading control. Results were reproduced in three independent experiments. (C, D) HEK293T cells were transfected with 700 ng vector, 200 ng BMP4 + 500 ng vector or 200 ng BMP4 + 500 ng BMP7R-GFlag. Equivalent volumes of conditioned media were subjected to trichloracetic acid precipitation or to immunoprecipitation (IP) with anti-Flag antibodies. Proteins were separated by electrophoresis under reducing conditions prior to immunoblotting (IB) with antibodies specific for BMP4 or Flag to detect BMP4 and BMP7 monomers. HEK293T cells were cultured for three hours in equivalent volumes of conditioned media from cells transfected with vector, BMP4 alone, or BMP4 + BMP7R-GFlag. Levels of pSmad1/5/8 were analyzed by immunoblot. Blots were reprobed for ß-actin as a loading control. Results were reproduced in two independent experiments.
DOI: https://doi.org/10.7554/eLife.48872.011

combination with a null allele of *Bmp2* or *Bmp4*. The heterodimer model predicts that removing a single copy of *Bmp2* or *Bmp4* will reduce the heterodimer pool, leading to a modest reduction in total BMP activity (illustrated at top of *Figure 5*). A further prediction is that the additional removal of a single copy of *Bmp7* ($Bmp2^{-/+};Bmp7^{-/+}$ or $Bmp4^{-/+};Bmp7^{-/+}$ compound mutants) will not lead to further depletion of the heterodimer pool, due to the ability of other class II BMPs to substitute for BMP7 in the heterodimer pool. Consistent with this prediction, *Bmp2, 4 or 7* null heterozygotes are adult viable and show mild skeletal defects that are not substantially worse in $Bmp2^{-/+};Bmp7^{-/+}$ or $Bmp4^{-/+};Bmp7^{-/+}$ compound heterozygotes (*Katagiri et al., 1998*). The heterodimer model predicts a different outcome in the case of $Bmp2^{-/+};Bmp7^{R-GFlag/+}$ or $Bmp4^{-/+};Bmp7^{R-GFlag/+}$ mice, since the BMP7R-GFlag protein sequesters a fraction of the available BMP2 and/or BMP4 in non-functional heterodimers (model, far right) such that a further reduction in *Bmp2* or *Bmp4* gene dosage will cause additional loss of the heterodimer pool (model, far right).

When $Bmp7^{R-GFlag/+}$ and $Bmp2^{-/+}$ mice were intercrossed, $Bmp2^{-/+};Bmp7^{R-GFlag/+}$ mutants were present at the predicted Mendelian ratios at E15.5, but were not recovered at weaning (*Table 4*). $Bmp2^{-/+}$ and $Bmp7^{R-GFlag/+}$ mice were indistinguishable from wild type littermates (*Figure 5A–C*),

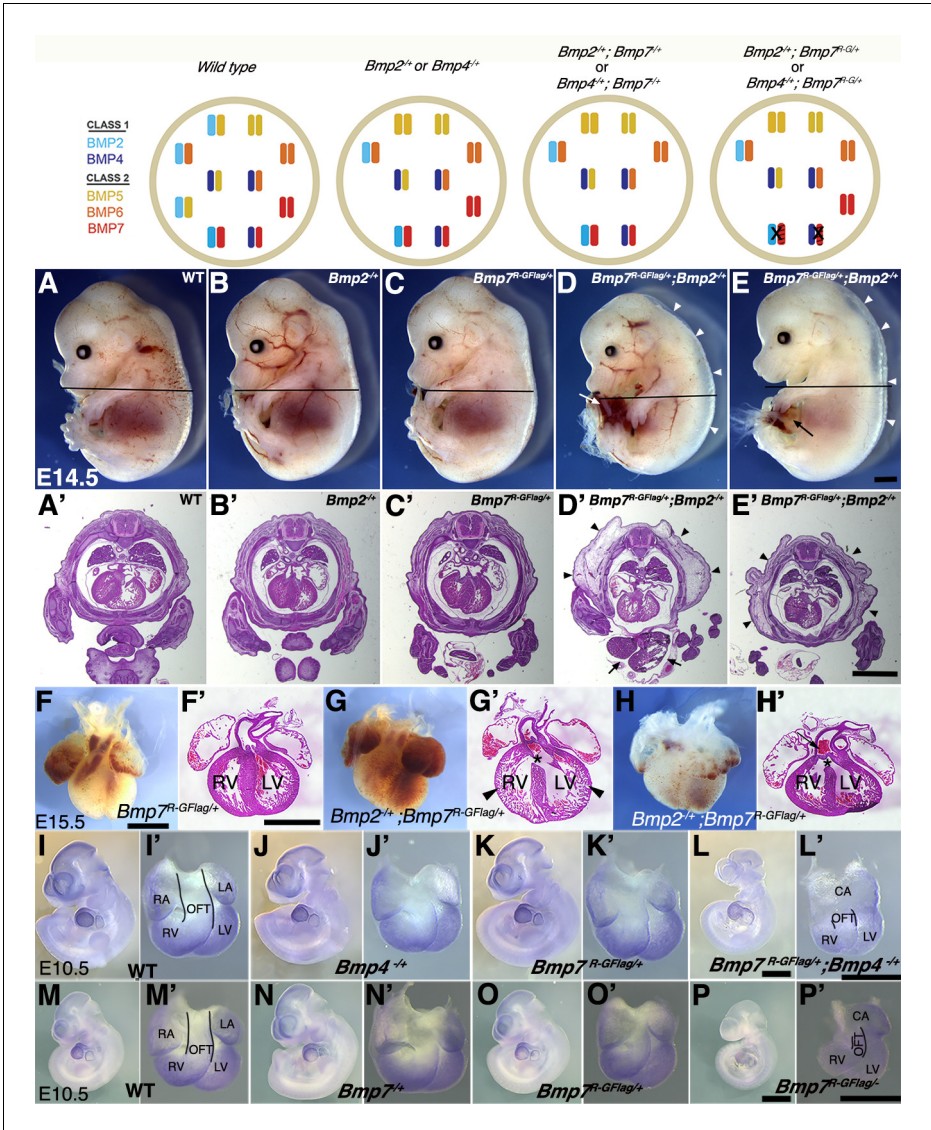

**Figure 5.** Analysis of compound heterozygotes shows that BMP2/7 and BMP4/7 heterodimers are required for early development. Schematic illustration of BMP activity in embryos carrying wild type or mutant alleles of *Bmp2, 4* and/or *7*. In the hypothetical model, class I/II heterodimers form preferentially. In addition, it is hypothesized that there is an excess of class II BMPs that form homodimers in the wild type condition but are available to form heterodimers to compensate for loss of any single class II BMP. In embryos lacking a single copy of *Bmp2* or *Bmp4*, activity contributed by the heterodimer pool is reduced but there is no further reduction in heterodimers when a single copy of *Bmp7* is also removed, due to redundancy with other Class II BMPs. A fraction of the heterodimer pool is inactivated in *Bmp7^{R-GFlag}* heterozygotes and additional removal of a single copy of *Bmp2* or *Bmp4* further depletes the heterodimer pool. (A–E') Photographs of E14.5 wild type and mutant littermates (A–E) and corresponding hematoxylin and eosin stained transverse sections through the abdomen (A'-E'; approximate position of section indicated by the black bar in A-E). Arrows indicated externalized viscera (D, D', E) and arrowheads denote peripheral edema (D, D', E, E'). (F–H') Ventral views of hearts dissected from E15.5 *Bmp7^{R-GFlag/+}* (F) or *Bmp7^{R-GFlag/+};Bmp2^{-/+}* embryos (G–H) and corresponding hematoxylin and eosin stained coronal sections (F'–H'). Asterisks denote VSDs (G', H'), arrow indicates abnormal positioning of the aorta exiting the right ventricle (H'), arrowheads indicate thin, non-compacted ventricular wall (G'). (I–P') Expression of *Nkx2.5* was analyzed by whole mount in situ hybridization in littermates generated by intercrossing *Bmp4^{-/+}* and *Bmp7^{R-GFlag/+}* (I–L') or *Bmp7^{-/+}* and *Bmp7^{R-GFlag/+}* mice (M–P'). Photographs of intact embryos at E10.5 (I–P) and photographs of hearts dissected from corresponding embryos (I'–P') are shown. RA; right atrium, LA; left atrium, CA; common atrium, RV; right ventricle, LV; left ventricle, OFT; outflow tract. The OFT is outlined in I', L', M' and P'. Scale bars in all panels correspond to 1 mm.

*Figure 5 continued on next page*

*Figure 5 continued*

DOI: https://doi.org/10.7554/eLife.48872.012

The following figure supplement is available for figure 5:

**Figure supplement 1.** Heart defects are present in *Bmp2$^{-/+}$;Bmp7$^{R-GFlag/+}$* embryos at E15.5 and are absent in *Bmp4$^{-/+}$;Bmp7$^{R-GFlag/+}$* and *Bmp7$^{R-GFlag/-}$* embryos at E9.5.

DOI: https://doi.org/10.7554/eLife.48872.013

whereas all (*n* = 10) E14.5 *Bmp2$^{-/+}$;Bmp7$^{R-GFlag/+}$* embryos showed peripheral edema (***Figure 5D–E, D'–E'***, arrowheads) along with defects in ventral body wall closure that ranged from umbilical hernia (***Figure 5E***, arrow, *n* = 4) to omphalocele, in which the liver and other visceral organs were externalized (***Figure 5D,D'***, arrows, *n* = 6). In addition, six of the ten compound heterozygotes were smaller than their littermates (***Figure 5D***). Peripheral edema is often associated with cardiovascular defects and thus we examined the hearts of E15.5 embryos. The heart from one *Bmp2$^{-/+}$;Bmp7$^{R-GFlag/+}$* embryo was much smaller than that of littermates and appeared atrophied (***Figure 5—figure supplement 1A,B***). Out of four *Bmp2$^{-/+}$;Bmp7$^{R-GFlag/+}$* hearts that were examined histologically, three showed ventricular septal defects (VSDs) (***Figure 5G',H'***, asterisks), and two showed defects in alignment of the aorta and pulmonary trunk (***Figure 5G',H'***) relative to wild type or single mutant siblings (***Figure 5F,F'***). In two compound mutants, the walls of the ventricles remained highly trabeculated and had not undergone compaction (***Figure 5G,G'***). An identical spectrum of ventral body wall, OFT and ventricular septal defects is observed in *Bmp2$^{-/+}$;Bmp4$^{-/+}$* compound mutants (***Goldman et al., 2009***; ***Uchimura et al., 2009***).

To assess whether BMP4/7 heterodimers are required for development, we intercrossed *Bmp7$^{R-GFlag/+}$* and *Bmp4$^{-/+}$* mice. Compound mutants were present at predicted Mendelian ratios at E9.5–11.5, but were not recovered at E12.5 or later (***Table 5***). *Bmp4$^{-/+}$;Bmp7$^{R-GFlag/+}$* embryos appeared grossly normal at E9.5, and expression of the BMP target gene *Nkx2.5* was intact in the heart (***Figure 5—figure supplement 1C–F'***). However, by E10.5 all eight embryos that were examined were smaller than littermates (***Figure 5I–L***) and showed defects in heart development (***Figure 5I–L'***). Specifically, whereas the hearts of wild type and single mutant siblings had morphologically distinguishable atria, ventricles, and OFT (***Figure 5I'–K'***), *Bmp4$^{-/+}$;Bmp7$^{R-GFlag/+}$* hearts had a common atrium (CA), and a smaller, malformed right ventricle and OFT (***Figure 5L'***). In addition, expression of the BMP target gene *Nkx2.5* was reduced in the hearts of all *Bmp4$^{-/+}$;Bmp7$^{R-GFlag/+}$* embryos (***Figure 5L'***) relative to littermates (***Figure 5I'–K'***).

To test whether other class II BMPs can compensate for loss of BMP7 in the heterodimer pool in *Bmp7$^{R-GFlag}$* heterozygotes, we intercrossed *Bmp7$^{R-GFlag/+}$* and *Bmp7$^{-/+}$* mice. *Bmp7$^{R-GFlag/-}$* mutants were present at predicted Mendelian ratios at E9.5–10.5 but were not recovered at E12.5 or later (***Table 6***). *Bmp7$^{R-GFlag/-}$* embryos appeared grossly normal (***Figure 5—figure supplement 1G–J***), and expression of the BMP target gene *Nkx2.5*, was intact in the heart at E9.5 (***Figure 5—figure supplement 1G'–J'***). By E10.5, however, all four *Bmp7$^{R-GFlag/-}$* embryos that were analyzed were smaller (***Figure 5P***) and their hearts showed a common atrium (CA), small right ventricle (RV), small malformed OFT and reduced expression of *Nkx2.5* (***Figure 5P'***) relative to siblings (***Figure 5M–O'***). By contrast, compound mutants heterozygous for the control *Bmp7$^{Flag}$* allele in combination with *Bmp2$^{-/+}$*, *Bmp4$^{-/+}$* or *Bmp7$^{-/+}$* were adult viable and showed no gross phenotypic defects (***Supplementary file 2***). Collectively, these findings suggest that heterodimers consisting of BMP7 together with BMP2 and/or BMP4 are essential for many early developmental processes including

**Table 4.** Progeny from *Bmp7$^{R-GFlag/+}$* and *Bmp2$^{-/+}$* intercrosses

| Age | Wildtype | Bmp2-/+ | Bmp7R-GFlag/+ | Bmp2-/+;Bmp7R-GFlag/+ | Total |
|---|---|---|---|---|---|
| P21** | 12 (50%) | 7 (29%) | 5 (29%) | 0 (0%) | 24 |
| E15.5-16.5 | 8 (26%) | 8 (26%) | 10 (32%) | 5 (16%) | 31 |
| E14.5 | 18 (37%) | 11 (23%) | 10 (20%) | 10 (20%) | 49 |

Data are presented as number (percent). Asterisks indicate that the observed frequency is significantly different than the expected frequency by Chi-square analysis (**P<0.01, *P<0.05).

DOI: https://doi.org/10.7554/eLife.48872.014

**Table 5.** Progeny from *Bmp7^R-GFlag/+* and *Bmp4^-/+* intercrosses

| Age | Wildtype | Bmp4-/+ | Bmp7R-GFlag/+ | Bmp4-/+;Bmp7R-GFlag/+ | Total |
|---|---|---|---|---|---|
| E12.5-14.5* | 9 (38%) | 8 (33%) | 7 (29%) | 0 (0%) | 24 |
| E11.5 | 5 (24%) | 4 (19%) | 5 (24%) | 7 (33%) | 21 |
| E10.5 | 5 (16%) | 7 (23%) | 11 (35%) | 8 (26%) | 31 |
| E9.5 | 14 (34%) | 11 (27%) | 7 (17%) | 9 (22%) | 41 |

Data are presented as number (percent). Asterisks indicate that the observed frequency is significantly different than the expected frequency by Chi-square analysis (*P<0.05).

DOI: https://doi.org/10.7554/eLife.48872.015

ventral body wall closure and formation of the heart. In addition, other class II BMPs cannot fully compensate for BMP7 in the heterodimer pool in the heart.

## Biochemical analysis reveals the existence of BMP4/BMP7 heterodimers in early embryos

To obtain biochemical evidence for heterodimer formation, BMP7 was immunoprecipitated from E11.5 *Bmp7^R-GFlag/+* protein lysates using antibodies directed against the Flag tag in the mature domain. Proteins in immunoprecipitates or in embryo lysates were separated by SDS-PAGE and immunoblots were probed with antibodies specific for the mature domain of BMP4 or for Flag as indicated below each panel. We have previously shown that on immunoblots of embryos lysates probed with the BMP4 antibody (*Figure 6A*, input), the band at ~55 kDa and the doublet at ~24 kDa correspond to the precursor protein and cleaved mature ligand (*Tilak et al., 2014*). In Flag immunoprecipitates, bands that co-migrate with the BMP4 precursor protein and the lower band in the cleaved mature BMP4 doublet were detected in lysates from *Bmp7^R-GFlag/+* embryos, but not from wild type littermates (*Figure 6A*, left panel). A band of the appropriate size for the BMP7R-GFlag precursor protein was detected in immunoprecipitates from *Bmp7^R-GFlag/+* embryos, but not from wild type littermates (*Figure 6A*, middle panel).

We also conducted co-immunoprecipitation assays using lysates from from E11.5 *Bmp^Flag* homozygotes. As shown in *Figure 6B*, when immunoblots of immunoprecipitates were probed with BMP4 antibodies, bands of the appropriate size for the BMP4 precursor protein and cleaved mature were detected in lysates from *Bmp7^Flag/Flag* embryos, but not from wild type littermates (left panel). Bands of the appropriate size for the BMP7Flag precursor protein and cleaved mature ligand were detected in immunoprecipitates from *Bmp7^Flag/Flag* embryos, but not from wild type littermates (*Figure 6B*, middle panel). The upper and lower panels in the Flag immunoblot represent two different exposures of the same blot (both exposures shown in *Figure 6—figure supplement 1*) because the longer exposure required to detect mature BMP7 obscured the precursor signal.

We also tried to detect BMP2/7, BMP5/7 and BMP6/7 heterodimers using co-immunoprecipitation assays but were unable to obtain antibodies sensitive enough to detect endogenous BMP2, BMP5 or BMP6 in vivo.

**Table 6.** Progeny from *Bmp7^R-GFlag/+* and *Bmp7^-/+* intercrosses

| Age | Wildtype | Bmp7-/+ | Bmp7R-GFlag/+ | Bmp7R-GFlag/- | Total |
|---|---|---|---|---|---|
| E12.5-14.5* | 7 (29%) | 9 (38%) | 8 (33%) | 0 (%) | 24 |
| E10.5 | 9 (37%) | 5 (21%) | 6 (25%) | 4 (17%) | 24 |
| E9.5 | 4 (17%) | 7 (31%) | 6 (26%) | 6 (26%) | 23 |

Data are presented as number (percent). Asterisks indicate that the observed frequency is significantly different than the expected frequency by Chi-square analysis (*P<0.05).

DOI: https://doi.org/10.7554/eLife.48872.016

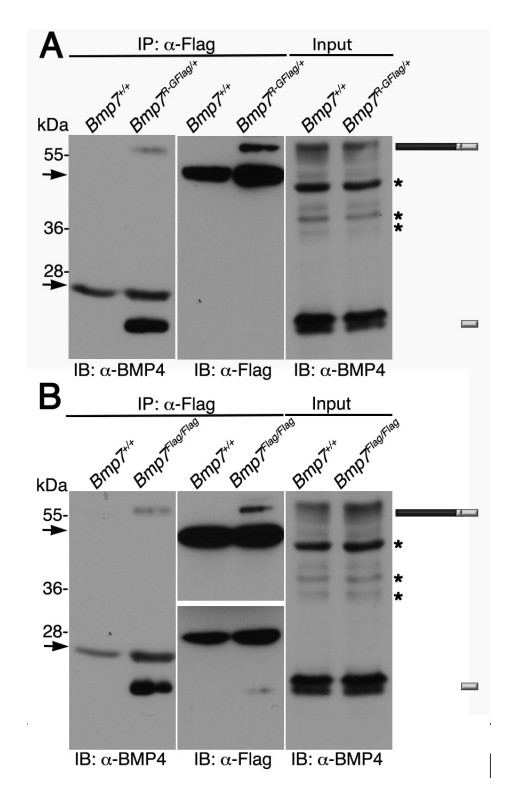

**Figure 6.** Endogenous BMP4 co-immunoprecipitates with BMP7. (**A**) Antibodies specific for the Flag-epitope tag were used to immunoprecipitate (IP) proteins from E11.5 *Bmp7*<sup>+/+</sup> or *Bmp7*<sup>R-GFlag/+</sup> lysates. Immunoblots (IB) of IPs or total protein (input) were probed with antibodies specific for BMP4 or Flag as indicated below each panel. (**B**) Antibodies specific for the Flag-epitope tag were used to immunoprecipitate (IP) proteins from E11.5 *Bmp7*<sup>+/+</sup> or *Bmp7*<sup>Flag/Flag</sup> lysates. Immunoblots (IB) of IPs or total protein (input) were probed with antibodies specific for BMP4 or Flag as indicated below each panel. (**A, B**) The position of precursor proteins and cleaved mature ligand is indicated on the right. Arrows denote bands corresponding to IgG heavy or light chains. The band at 28 kDa on the middle panel is a non-specific band detected by HRP-conjugated Flag antibody. Asterisks mark bands that are considered non-specific because only the bands marked as precursor and ligand show decreased intensity on BMP4 immunoblots of lysates from *Bmp4*<sup>+/-</sup> relative to *Bmp4*<sup>+/+</sup> embryos (*Tilak et al., 2014*) and unpublished results). Results were reproduced in three independent experiments.

DOI: https://doi.org/10.7554/eLife.48872.017

The following figure supplement is available for figure 6:

**Figure supplement 1.** Short and long exposure of Flag immunoblot.

DOI: https://doi.org/10.7554/eLife.48872.018

## Discussion

Previous studies have shown that heterodimers composed of BMP7 together with BMP2 or BMP4 have a higher specific activity than individual homodimers in specific in vitro assays, but it was unknown whether, or to what extent, endogenous class I/II BMP heterodimers are required for mammalian development. The current studies demonstrate that BMP2/7 and/or BMP4/7 heterodimers are the predominant functional signaling ligand in many tissues of early mouse embryos.

BMP activity is intact in the eye field of *Bmp7* null mutants at E9.5, but is absent in *Bmp7*<sup>R-GFlag</sup> homozygotes, suggesting that BMP7-containing heterodimers play an essential role in early inductive events in the eye. *Bmp4* and *Bmp7* are co-expressed in head surface ectoderm at the time of lens placode induction (E9), and both have been implicated in this process (*Dudley and Robertson, 1997*; *Furuta and Hogan, 1998*; *Wawersik et al., 1999*). Analysis of embryos engineered to express *Bmp4* or *Bmp6* from the *Bmp7* allele demonstrates that BMP6 can rescue eye defects in *Bmp7* null mutants, whereas BMP4 cannot (*Oxburgh et al., 2005*). This finding is consistent with the possibility that the higher specific activity of endogenous BMP4/7 heterodimers is essential to generate sufficient BMP activity for lens induction. In this scenario, another class II BMP (BMP6) could substitute for BMP7 to rescue heterodimer formation, whereas a class I BMP (BMP4) could not.

The role of BMP4/7 heterodimers in eye development is likely conserved in humans since *Bmp4* or *Bmp7* are co-expressed in the developing human eye, and mutations in either gene are associated with anophthalmia, microphthalmia and chorioretinal coloboma (*Bakrania et al., 2008*; *Wyatt et al., 2010*). Point mutations within the prodomain of BMP4 or BMP7 are also associated with an overlapping spectrum of brain and palate abnormalities (*Bakrania et al., 2008*; *Suzuki et al., 2009*; *Wyatt et al., 2010*; *Reis et al., 2011*), and several of these lead to single amino acid substitutions within short regions of the prodomain that are highly conserved between BMP4 and BMP7. We have shown that the prodomain of BMP4 is both necessary and sufficient to generate heterodimeric BMP4/7 ligands (*Neugebauer et al., 2015*), raising the possibility that the amino acid substitutions interfere with heterodimer formation.

The heart defects observed in *Bmp7*<sup>R-GFlag</sup> homozygotes appear earlier (by E9.5) but otherwise phenocopy those in *Bmp7*<sup>R-GFlag/+</sup>;*Bmp4*<sup>-/+</sup>

and $Bmp7^{R-GFlag/-}$ mutants. This suggests that BMP4/7 heterodimers play essential roles in early stages of heart development that cannot be compensated for by other BMP family members. BMP4 is essential for septation of the ventricles, atrioventricular canal and outflow tract, as well as for valve formation and remodeling of the branchial arch arteries (*Jiao et al., 2003*; *Liu et al., 2004*; *McCulley et al., 2008*). Although BMP2 is not able to compensate for BMP4 during early heart development, our finding that $Bmp7^{R-GFlag/+};Bmp2^{-/+}$ compound heterozygotes have defects in the heart and in ventral body wall closure that phenocopy those observed in *Bmp2;Bmp4* compound heterozygotes (*Goldman et al., 2009*; *Uchimura et al., 2009*) suggest that BMP2 and BMP4 function redundantly as heterodimeric partners with BMP7 later in development. While *Bmp7* null mutants do not show defects in heart development, conditional deletion of both *Bmp7* and *Bmp4* from progenitors of the secondary heart field leads to persistent truncus arteriosus (*Bai et al., 2013*). Collectively, our results raise the possibility that these defects are due in part to reduction in the BMP4/7 and BMP2/7 heterodimer pool, rather than the loss of functionally redundant BMP4 and BMP7 homodimers.

Our observations that endogenous BMP activity is intact in the roof plate of the spinal cord in *Bmp7* null mutants at E9.5, but is reduced in $Bmp7^{R-GFlag}$ homozygotes suggest that heterodimers containing BMP7 are the physiologically relevant ligand(s) that are secreted from the surface ectoderm to induce the roof plate. *Bmp2, 4* and *7* are co-expressed in surface ectoderm overlying the neural tube by E8.5, whereas *Bmp5* and *Bmp6* are not expressed in this tissue at this stage (*Dudley and Robertson, 1997*; *Danesh et al., 2009*). *Bmp5;Bmp6* and *Bmp5;Bmp7* double mutants show grossly normal dorsoventral patterning of the spinal cord (*Solloway and Robertson, 1999*; *Kim et al., 2001*), suggesting that, in the absence of class II BMPs, BMP2 and BMP4 can instead form homodimers that are sufficient for roof plate induction.

Defects in *Bmp2* or *Bmp4* null mutants overlap with, but are more severe than those observed in $Bmp7^{R-GFlag}$ mutants. *Bmp2* null mutants die between day 7 and 10.5 of gestation due to failure of amnion/chorion formation and an abnormal heart forms in the exocoelomic cavity, rather than in its normal location in the amniotic cavity (*Zhang and Bradley, 1996*). Most *Bmp4* null mutants die prior to E8 and show little or no mesoderm formation. Those that survive to E9.5 are developmentally retarded, have small or disorganized posterior structures, smaller limb buds and little or no blood (*Winnier et al., 1995*). $Bmp7^{R-GFlag}$ mutants are developmentally delayed, have heart defects and smaller limb buds but do not show mislocalization of the heart nor defects in the amnion or chorion. BMP5 and/or BMP6 may function redundantly with BMP7 to form heterodimers with type I ligands in some tissues, and type I ligands may signal as homodimers in others, which would account for the discordance in phenotypes.

One caveat to our conclusion that the phenotypic defects in $Bmp7^{R-GFlag}$ mutants are caused by loss of class I/II heterodimers is the possibility that BMP7R-GFlag forms inactive heterodimers with BMP5 and/or BMP6, effectively creating a double null mutant. While inactivation of BMP5 and/or BMP6 may indeed contribute to reduction in BMP activity in some tissues, including the heart, it cannot fully account for the defects observed in $Bmp7^{R-GFlag}$ homozygotes since they do not phenocopy those observed in *Bmp5;Bmp7* or *Bmp6;Bmp7* double mutants (*Solloway and Robertson, 1999*; *Kim et al., 2001*).

The *Drosophila* BMP5-8 orthologs Screw and Glass bottom boat (GBB) undergo proteolytic processing at sites within the prodomain, in addition to cleavage of the site adjacent to the mature domain (*Akiyama et al., 2012*; *Fritsch et al., 2012*; *Künnapuu et al., 2014*). In the case of GBB, cleavage of the prodomain site alone is sufficient to generate a bioactive ligand that signals at longer range than the conventional small ligand (*Akiyama et al., 2012*). Putative upstream PC consensus motifs can be identified within the prodomain of mammalian BMP7 (*Akiyama et al., 2012*), but the early lethality of $Bmp7^{R-GFlag}$ homozygotes demonstrates that cleavage at cryptic sites within the prodomain is not sufficient to generate functional BMP7 ligands that can support development. Furthermore, we are unable to detect BMP7 fragments generated by cleavage(s) at sites other than the previously identified PC motif in lysates from mouse embryos (current studies), *Xenopus* embryos or mammalian cells, or when BMP7 is cleaved by recombinant furin in vitro (*Sopory et al., 2006*; *Neugebauer et al., 2015*). However, it remains possible that the cryptic sites are cleaved in select tissues.

The endogenous BMP4 precursor is efficiently cleaved when dimerized with BMP7R-GFlag, raising questions as to how the mutant precursor blocks the function of wild type partners. BMP7

homodimers (*Jones et al., 1994*) and BMP4/7 heterodimers (*Neugebauer et al., 2015*) are secreted as a stable complex consisting of the cleaved mature ligand noncovalently associated with both pro-peptides. Structural studies have shown that homodimeric precursors of ActivinA and TGFß adopt a crossed-arm, domain-swapped configuration in which the amino-terminal part of the prodomain is in close contact with the ligand domain derived from the same precursor monomer but the bulk of the prodomain crosses over to interact with the mature domain derived from the second monomer (*Wang et al., 2016*; *Zhao et al., 2018*). If this structural paradigm holds for heterodimers as well, then the BMP7 prodomain interacts with the BMP4 mature domain. Previous studies have shown that the BMP7 prodomain remains non-covalently associated with mature BMP7 homodimers, and that Type II BMP receptors must displace the cleaved BMP7 prodomain to initiate signaling (*Sengle et al., 2008*). Because the uncleaved BMP7R-GFlag prodomain cannot be displaced from the ligand by the Type II receptors, the heterodimeric ligand is most likely unable to assemble an active receptor complex.

BMP4 and BMP7 preferentially form heterodimers rather than either homodimer when the two molecules are co-expressed in the same cell in *Xenopus* (*Neugebauer et al., 2015*). The current results suggest that this is a common theme for class I and class II BMPs that are expressed in overlapping patterns. However, BMP2 and BMP7 form equivalent amounts of heterodimer and each homodimer when expressed in zebrafish (*Little and Mullins, 2009*) suggesting that the relative abundance of heterodimers and homodimers may differ depending on tissue and organism. Thus, the functional importance of heterodimers versus homodimers is likely to vary widely among different tissues and developmental stages. Additional biochemical and phenotypic analysis will be required to sort out which ligands are used in which tissues.

## Materials and methods

### Key resources table

| Reagent type (species) or resource | Designation | Source or reference | Identifiers | Additional information |
|---|---|---|---|---|
| Genetic reagent (*M. musculus*) | *Bmp4-/+* | PMID: 10049358 | RRID:MGI:2664348 | Dr. Brigid Hogan (Duke University) |
| Genetic reagent (*M. musculus*) | *BRE-LacZ* | PMID: 15331632 | | Dr. Christine Mummery (Leiden University) |
| Genetic reagent (*M. musculus*) | *Bmp2-/+* | PMID: 8898212 | RRID:MGI:2658703 | Dr. Yuji Mishina (University of Michigan) |
| Genetic reagent (*M. musculus*) | *Bmp7-/+* | PMID: 9693150 | RRID:IMSR_EM:02513 | Dr. Elizabeth Robertson (University of Oxford) |
| Genetic reagent (*M. musculus*) | *Bmp7flox/flox* | PMID: 22219353 | RRID:MGI:5312875 | Dr. James Martin (Baylor University) |
| Genetic reagent (*M. musculus*) | *CMV-CRE* | PMID: 8559668 | RRID:IMSR_JAX:006054 | University of Utah, Transgenic Mouse Facility |
| Genetic reagent (*M. musculus*) | *Bmp7R-GFlag* | This paper | | Generated using gene targeting technology |
| Genetic reagent (*M. musculus*) | *Bmp7Flag* | This paper | | Generated using CRISPR-Cas9 technology |
| Transfected construct (Synthesized) | CS2+BMP4 | PMID: 15356272 | | Catherine Degnin (Oregon Health and Science University) |
| Transfected construct (Synthesized) | CS2+BMP7R-GFlag | This paper | | PCR used to insert Flag tag in cDNA |
| Antibody | Rabbit polyclonal anti-pSmad1/5/8 | Cell Signaling | Cat. #9511S | WB (1:1000), IHC (1:500) |
| Antibody | Rabbit polyclonal anti-Alexa Fluor 488 | Invitrogen | Cat. #11008 | IHC (1:500) |

*Continued on next page*

*Continued*

| Reagent type (species) or resource | Designation | Source or reference | Identifiers | Additional information |
|---|---|---|---|---|
| Antibody | Mouse monoclonal anti-BMP4 | Santa Cruz | Cat. # sc-12721 | WB (1:1000) |
| Antibody | Mouse monoclonal anti-Flag M2 | Sigma | Cat. # F1804 | WB (1:1000) |
| Antibody | HRP-conjugated mouse monoclonal Flag M2 | Sigma | Cat. # A8592 | WB (1:5,000) |
| Antibody | Rabbit polyclonal anti-beta actin | AbCam | Cat. # ab8227 | WB (1:10,000) |
| Antibody | HRP-conjugated anti-rabbit polyclonal IgG | Jackson ImmunoResearch | Cat. # 111-035-144 | WB (1:10,000) |
| Antibody | HRP-conjugated anti-mouse polyclonal IgG2b | Jackson ImmunoResearch | Cat. # 115-035-207 | WB (1:10,000) |
| Antibody | Anti-Flag mouse monoclonal M2 Agarose | Sigma | Cat. # A2220 | IP (1:500) |
| Cell line (*H. sapiens*) | HEK293T | American Type Culture Collection | Cat. # CRL-11268, RRID:CVCL_0045 | |
| Commercial assay or kit | *lmx1A* probe set, HCR amplification and buffer | Molecular Instruments | | |
| Commercial assay or kit | ECL Prime Western Kit | Fisher | Cat. # 45-010-090 | |
| Commercial assay or kit | BCA Protein Assay Kit | Fisher | Cat. # 23225 | |
| chemical compound, drug | Halt protein and phosphatase inhibitor | Fisher | Cat. # 78442 | |

## Mouse strains

Animal procedures followed protocols approved by the University of Utah Institutional Animal Care and Use Committees. $Bmp4^{LacZ/+}$, $Bmp2^{-/+}$ and BRE-LacZ mice were obtained from Dr. B Hogan (Duke University), Dr. Y. Mishina (University of Michigan) and Dr. C Mummery (Leiden University), respectively. $Bmp7^{tm2Rob}$ mice were obtained from Dr. E Robertson (Cambridge University) and were used for all phenotypic analysis. $Bmp7^{flox/flox}$ mice were obtained from Dr. J Martin (Baylor) and were crossed to CMV-cre mice to generate a null allele for analysis of BMP activity in BRE-LacZ crosses.

## Generation and genotyping of mice

The targeting vector used to generate $Bmp7^{R-GFlagNeo}$ mice was constructed from BAC clone bMQ298P20 purchased from Source Bioscience. This targeting construct (illustrated in *Figure 1—figure supplement 1*) includes: (a) sequence encoding an in frame Flag epitope tag within the mature domain located 24 amino acids downstream of the cleavage site (-EALRMDYKDDDDKAS-VAG-; Flag epitope underlined), (b) two point mutations in exon four that introduce an arginine to glycine amino acid change at the S2 cleavage site (RISR-RISG) and a new BamHI site, and (d) a neomycin selectable marker flanked by loxP sites upstream of exon 4. Linearized vector was electroporated into R1 ES cells and homologous recombinants were selected with G418 and gancyclovir. Correctly targeted ES cell clones were identified by Southern analysis using probes derived from genomic sequences located both internal and external to the targeting vector. Positive clones were expanded and mutations and epitope tag sequences were verified by sequencing DNA fragments PCR-amplified from genomic DNA. Heterozygous ES cells were injected into C57BL/6J blastocysts, and the resulting chimeras were mated with C57BL/6J females to obtain $Bmp7^{R-GFlagNeo}$

heterozygotes. Two independent mouse lines for each strain were mated to Cre deleter mice (*Schwenk et al., 1995*) to remove the neomycin gene.

*Bmp7^Flag^* mice were generated using CRISPR-Cas9 mutagenesis as described (*Qin et al., 2016*). sgRNA RNA (5'-CTCGGACCTACCTGCCACAC-3') was synthesized by in vitro transcription of an oligo-based template and was injected into C57BL/6J zygotes together with a single stranded donor DNA repair template (5'-CGCAGCCAGAATCGCTCCAAGACGCCAAAGAACCAAGAGGCCC TGAGGATGGACTACAAAGACGATGACGATAAAGCtAGcGTGGCAGgtaggtccgagcagctggaggggac-cagctcattgcagatgctt-3'; sequence encoding FLAG epitope underlined) and Cas9 protein. G0 founders were crossed to C57BL/6J females to obtain heterozygotes. DNA fragments PCR-amplified from genomic DNA were sequenced to verify the presence of the epitope tag and absence of other sequence changes. Genotypes were determined by PCR amplification of tail DNA using primers that anneal to sequence immediately surrounding the Flag epitope tag (5' primer: 5'- CAAG TTGGCAGGCCTGAT-3' and 3' primer: 5'- AAAGACACGTCCCAGGTCAC-3') under the following conditions: 94°C for 30 s, 60°C for 30 s, 72°C for 30 s, 35 cycles.

## Immunostaining, in situ hybridization and ß-galactosidase staining

For phosphoSmad immunostaining, E9.5 embryos were fixed in 4% paraformaldehyde in PBS at 4°C for one hour, incubated overnight in 30% sucrose in PBS at 4°C and then embedded in OCT (Tissue-Tek). 10 µm cryosections were incubated overnight at 4°C with an antiphosphoSmad1/5/8 antibody (1:500; Cell Signaling 9511S) in PBS with 5% goat serum and 0.1% Triton X-100. Staining was visualized using anti-rabbit Alexa Fluor 488-conjugated secondary antibody (1:500; Molecular Probes). Embryos were processed for in situ hybridization with digoxigenin-labeled *Nkx2.5* riboprobes as described previously (*Wilkinson and Nieto, 1993*). Quantitative in situ HCR was performed as described (*Trivedi et al., 2018*) using a *lmx1a* DNA probe set, a DNA HCR amplifier and hybridization, wash and amplification buffers purchased from Molecular Instruments. Whole mount mouse embryos were processed for in situ HCR as described (*Choi et al., 2016*). ß-galactosidase staining of *BRE-LacZ* embryos was performed as described (*Lawson et al., 1999*). Investigators were blinded to genotype until after morphology and/or staining intensity had been documented.

## Histology

Isolated embryos or dissected hearts were fixed in 4% paraformaldehyde in PBS, dehydrated and embedded in paraffin. Sections (10 µm) were stained with Hematoxylin and eosin.

## Transient transfection and western blot analysis of cultured cells

HEK293T cells (authenticated at the University of Utah DNA sequencing core and tested for mycoplasma in the lab) were plated on 10 cm culture dishes and transfected with 500 ng of DNA encoding BMP4, BMP7R-GFlag or empty vector (pCS2+) for the experiments shown in *Figure 4B*. Cells were cultured for one day in serum containing media and then cultured for one additional day in serum free media before collecting conditioned media. HEK293T cells were transfected with 200 ng of DNA encoding BMP4 + 500 ng pCS2+, 200 ng BMP4 + 500 ng of BMP7R-GFlag or 700 ng pCS2 + for the experiments shown in *Figure 4D*. Cells were cultured for one day in serum containing media and then cultured for one additional day in serum free media before collecting conditioned media. Equivalent amounts of media were collected for immunoblotting or immunoprecipitation followed by immunoblotting. HEK293T cells were incubated with equivalent volumes of conditioned- or control media then lysed and used for immunoblot analysis. Proteins were separated by electrophoresis on 10% or 12% SDS-polyacrylamide gels and transferred to PVDF membranes that were probed with anti-pSmad1/5/8 (Cell Signaling 9511S), anti-BMP4 (Santa Cruz sc12721), anti-Flag M2 (Sigma F1804), anti-HRP-conjugated Flag M2 (Sigma A8592) or anti-ßactin (Abcam ab8229) primary antibodies followed by HRP-conjugated anti-rabbit IgG or HRP-conjugated anti-mouse IgG2b heavy chain specific (Jackson ImmunoResearch) secondary antibodies. Immunoreactive proteins were visualized using an ECL prime kit (GE HealthCare).

## Co-immunoprecipitation assays

Embryos were dissected from pregnant females at E11.5, homogenized in IP lysis buffer (150 mM NaCl, 20 mM Tris-Cl pH 7.5, 1 mM EDTA, 1% Sodium deoxycholate, 1% NP40, 1X protease inhibitor

(Thermo Scientific)) and protein concentration was measured using a BCA kit (Thermo Scientific). 1 mg of embryo lysate or 400 μl of conditioned media from HEK293T cells was diluted to 1 ml with IP lysis buffer pre-cleared by incubating with 100 μl pre-cleared protein A/G agarose for 2 hr at 4°C. Samples were spun for 5 min in a microfuge and 950 μl of supernatant was transferred to a new tube and incubated with agarose beads-conjugated to anti-Flag antibody (1:500; Sigma) overnight at 4°C, followed by three 10 min washes in IP buffer. Samples were spun for 5 s in a microfuge; supernatant was discarded and proteins were recovered in 40 μl 2X Laemmli sample buffer (BioRad) by boiling for 5 min prior to SDS-PAGE and immunoblot analysis.

### Skeletal preparations
Skeletal staining was performed as described (*Hogan et al., 1994*).

## Acknowledgements

We thank Anne E Martin for generating the schematic illustrations in *Figure 1* and *Figure 5*, Isabelle Cooperstein for managing the mouse colony and performing in situ HCR shown in *Figure 3* and Chris Gregg, Suzi Mansour and Rich Dorsky for helpful comments on the manuscript. This work was supported by the National Institutes of Health (RO1HD037976 to JLC, T32DK007115 to JMN and T32HD007491 to AMN). This work utilized DNA, peptide, transgenic mouse and imaging shared resources supported by the Huntsman Cancer foundation and the National Cancer Institute of the NIH (P30CA042014) and the mutation generation and detection core supported in part by a grant from the National Institute of Diabetes and Digestive and Kidney Diseases (U54DK110858). The content is solely the responsibility of the authors and does not represent the official views of the NIH.

## Additional information

### Funding

| Funder | Grant reference number | Author |
|---|---|---|
| National Institutes of Health | RO1HD037976 | Jan L Christian |
| National Institutes of Health | T32DK007115 | Judith Neugebauer |
| National Institutes of Health | T32HD007491 | Autumn McKnite |
| National Institute of Child Health and Human Development | R01HD067473 | Jan L Christian |

The funders had no role in study design, data collection and interpretation, or the decision to submit the work for publication.

### Author contributions
Hyung-Seok Kim, Conceptualization, Data curation, Formal analysis, Validation, Methodology, Writing—review and editing; Judith Neugebauer, Conceptualization, Formal analysis, Visualization; Autumn McKnite, Formal analysis, Investigation, Visualization; Anup Tilak, Formal analysis, Visualization; Jan L Christian, Conceptualization, Supervision, Funding acquisition, Validation, Writing—original draft, Project administration, Writing—review and editing

### Author ORCIDs
Jan L Christian https://orcid.org/0000-0003-3812-3658

### Ethics
Animal experimentation: This study was performed in strict accordance with the recommendations in the Guide for the Care and Use of Laboratory Animals of the National Institutes of Health. All animal procedures followed protocols approved by the University of Utah Institutional Animal Care and Use Committee (protocol #17-03007).

Decision letter and Author response
Decision letter https://doi.org/10.7554/eLife.48872.023
Author response https://doi.org/10.7554/eLife.48872.024

## Additional files

### Supplementary files

• Supplementary file 1. Progeny from $Bmp7^{R-GFlag/+}$ and $Bmp7^{+/+}$ intercrosses. Numbers and percent of animal of each genotype at P28.
DOI: https://doi.org/10.7554/eLife.48872.019

• Supplementary file 2. Progeny from $Bmp7^{Flag/+}$ and $Bmp2^{-/+}$, $Bmp4^{-/+}$, and $Bmp7^{-/+}$ intercrosses. (A) Progeny from $Bmp7^{Flag/+}$ and $Bmp2^{-/+}$ intercrosses. (B) Progeny from $Bmp7^{Flag/+}$ and $Bmp4^{-/+}$ intercrosses. (C) Progeny from $Bmp7^{Flag/+}$ and $Bmp7^{-/+}$ intercrosses.
DOI: https://doi.org/10.7554/eLife.48872.020

• Transparent reporting form
DOI: https://doi.org/10.7554/eLife.48872.021

### Data availability

All data generated and analyzed during this study are included in the manuscript and supporting files.

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
