## [Decision Letter]

Thank you for submitting your article "BMP7 functions predominantly as a heterodimer with BMP2 or BMP4 during mammalian embryogenesis" for consideration by *eLife*. Your article has been reviewed by three peer reviewers, and the evaluation has been overseen by Marianne Bronner as the Senior and Reviewing Editor. The reviewers have opted to remain anonymous.

The reviewers have discussed the reviews with one another and the Reviewing Editor has drafted this decision to help you prepare a revised submission.

Summary:

This paper addresses whether TGF-β/BMP fligands function predominantly as homodimers or heterodimers in the mouse. To this end, they generate mouse knockin for a mutation in BMP7 that prevents furin-dependent cleavage of the pro-domain from the mature domain during the secretion process, thus abolishing function of all BMP7 homodimers, plus any heterodimers that it forms. The authors convincingly show that BMP7 heterodimerizes with BMP2 and BMP4 and that these heterodimers are responsible for signaling in a wide variety of tissues. The data are of high quality and the paper will make an important contribution to the literature once appropriately revised in accordance with the reviewers' comments.

Essential revisions:

The authors should work out better what the mechanism is by which the non-cleavable BMP is functioning and determine the class I and class II BMPs with which BMP7 is heterodimerizing. They also need to rule out a general dominant negative effect that goes beyond BMP7. In addition, comparing their results with other class I BMP null mutants will be informative. Finally, it is important to replace Figure 6 for the reasons described in the full reviews below.

Reviewer #1:

The paper "BMP7 functions predominantly as a heterodimer with BMP2 and BMP4 during mammalian embryogenesis" is of high quality, with the experiments well thought out and the data strongly supporting the conclusions. The paper is also very well written and the figures are well designed (with exceptions below). The authors mention in the Discussion that the BMP7R-Gflag ligand causes defects in the roof plate patterning, and as BMP5 and BMP6 are not expressed in this tissue during this stage, BMP7 containing heterodimers must be the main ligand. However, the authors mention that Bmp7 null, Bmp5;6, and Bmp5;7 mutants all have normally patterned neural tubes. Thus, this suggests that BMP7R-Gflag is disrupting processes that do not necessarily require endogenous BMP7, or BMP7R-Gflag is inherently dominant negative. But I may be missing something here. Also, the authors imply that the class I ligand functions mainly as heterodimers with the class II ligands, but they make few direct comparisons between the phenotypes of the non-cleavable BMP7R-Gflag and the type I ligand mutants, which could provide additional evidence consistent with heterodimers.

Reviewer #2:

This paper addresses an important issue in the TGF-β/BMP field, which is whether these ligands function predominantly as homodimers, or as heterodimers. In fish, it is clear that BMP2 and BMP7 function as obligate heterodimers, at least at early stages of development, but this issue has not been rigorously addressed in mammals. The authors have addressed this issue in mouse, by generating a mouse knockin for a mutation in BMP7 that would prevent furin-dependent cleavage of the pro-domain from the mature domain during the secretion process. This mutation would be expected to abolish function of all BMP7 homodimers, plus any heterodimers that it forms. The authors show clearly that this mutant has a stronger phenotype than a BMP7 null. They rationalize this by assuming that in the case of the BMP7 null, other class II BMPs would be able to compensate as both homodimers and heterodimers, whilst in the case of the non-cleavable BMP7 mutant, it would essentially act as a dominant negative and bind BMP2 and BMP4, locking them into inactive complexes, as well as abolishing all function of BMP7.

I think the quality of the data is high and the mouse results are very clear. I think that the authors have convincingly shown that BMP7 heterodimerizes with BMP2 and BMP4 and that these heterodimers are responsible for signaling in a wide variety of tissues.

The only weakness of the paper is in understanding how the BMP7 R-G mutation is functioning. This is important, as it influences the interpretation of the data. One possible concern is that it could be having a dominant negative effect on furin activity per se, and thus affecting the cleavage of multiple ligands, beyond BMP7. The experiment that would apparently rule this out is the one in Figure 6, where they show that BMP4 is cleaved normally. However, the authors should provide proof that the band they detect, immunoprecipitating with the uncleaved BMP7 really is mature BMP4.

In addition, I think the authors need to nail down the mechanism whereby the uncleavable BMP7 is inhibiting function. This could be easily be done in a tissue culture system, with overexpressed ligands if necessary. I also think it is important to understand the entire range of BMPs that BMP7 is heterodimerizing with in vivo, as this affects the data interpretation. Is this exclusively class I, or can BMP7 also dimerize with other class II BMPs? This should be addressed.

Reviewer #3:

This manuscript addresses the important question of the extent to which different members of the BMP family form heterodimers with each other in vivo and whether such heterodimeric forms deliver unique functions or are potentially more potent in particular biological contexts than homodimeric BMPs. This manuscript nicely demonstrates that heterodimeric BMPs are indeed generated in vivo in developing mouse embryos. Because of the general importance of the question, I am overall supportive of publication of the manuscript. However, the current paper has several issues that need to be addressed before acceptance.

1) Figure 6B should be replaced with a better figure. Why is the middle portion of the control western blot using anti-Flag antibody missing? Why are there three heavy and light chain bands in panel B? There should be only two bands. What are the additional non-specific bands marked by asterisks? Also, panels A and B need to be referred to in the figure legend.

2) Because the result in Figure 5 indicates that BMP7 and BMP2 form heterodimers, it would be useful to also show evidence of BMP7/BMP2 heterodimers in developing embryos by western blot, as was done in Figure 6 for BMP4/BMP7 heterodimers. Is there a reason why this cannot be easily accomplished?

---

## [Author Response]

Reviewer #1:

The paper "BMP7 functions predominantly as a heterodimer with BMP2 and BMP4 during mammalian embryogenesis" is of high quality, with the experiments well thought out and the data strongly supporting the conclusions. The paper is also very well written and the figures are well designed (with exceptions below). The authors mention in the Discussion that the BMP7R-Gflag ligand causes defects in the roof plate patterning, and as BMP5 and BMP6 are not expressed in this tissue during this stage, BMP7 containing heterodimers must be the main ligand.However, the authors mention that Bmp7 null, Bmp5;6, and Bmp5;7 mutants all have normally patterned neural tubes. Thus, this suggests that BMP7R-Gflag is disrupting processes that do not necessarily require endogenous BMP7, or BMP7R-Gflag is inherently dominant negative. But I may be missing something here.

We had discussed this point in the lab but failed to include it in the manuscript. Thank you for bringing it to our attention. Our interpretation is that reduction in pSmad1 and *lmx1a* in *Bmp7^RGFlag^* homozygotes suggests Bmp7R-GFlag disables endogenous heterodimers that normally contribute to roof plate induction. However, in the absence of *Bmp7, Bmp2* and *Bmp4* must instead form homodimers that are sufficient for roof plate induction. We have revised the text to clarify this (Discussion, fifth paragraph).

Also, the authors imply that the class I ligand functions mainly as heterodimers with the class II ligands, but they make few direct comparisons between the phenotypes of the non-cleavable BMP7R-Gflag and the type I ligand mutants, which could provide additional evidence consistent with heterodimers.

We have added a paragraph to the Discussion comparing the mutant phenotypes (Discussion, sixth paragraph).

Reviewer #2:

[…] The only weakness of the paper is in understanding how the BMP7 R-G mutation is functioning. This is important, as it influences the interpretation of the data. One possible concern is that it could be having a dominant negative effect on furin activity per se, and thus affecting the cleavage of multiple ligands, beyond BMP7. The experiment that would apparently rule this out is the one in Figure 6, where they show that BMP4 is cleaved normally. However, the authors should provide proof that the band they detect, immunoprecipitating with the uncleaved BMP7 really is mature BMP4.

We have repeated the co-immunoprecipitation experiment (Figure 6A) to give better separation between the IgG band and the cleaved BMP4 ligand band. The evidence that this band really is cleaved mature BMP4 is that it migrates at the same position as mature BMP4 in embryo lysates (Figure 6A, last two lanes). In a previous publication, we showed that the intensity of the precursor and ligand band(s) recognized by this antibody decrease by half in *Bmp4* null heterozygotes and we have expanded the description of the results to clarify this (subsection “Biochemical analysis reveals the existence of BMP4/BMP7 heterodimers in early embryos”, first paragraph and Figure 6 legend). Please also see the new data presented in Figure 4D, which confirms that even highly overexpressed BMP7R-GFlag does not have a dominant negative effect on proprotein convertase activity per se.

In addition, I think the authors need to nail down the mechanism whereby the uncleavable BMP7 is inhibiting function. This could be easily be done in a tissue culture system, with overexpressed ligands if necessary.

We have added new data (Figure 4C, D) showing that heterodimers of uncleaved BMP7R-GFlag and cleaved BMP4 are secreted but are not able, or less able to activate pSmad1, thereby supporting a model in which they cannot assemble an active receptor complex outside of cells. We have expanded the description of the evidence for this mechanism in the Results (subsection “Bmp7R-GFlag homodimers do not act outside of cells to block BMP activity”, last paragraph) and Discussion (Discussion, ninth paragraph).

*I also think it is important to understand the entire range of BMPs that BMP7 is heterodimerizing with* in vivo*, as this affects the data interpretation. Is this exclusively class I, or can BMP7 also dimerize with other class II BMPs? This should be addressed.*

We attempted BMP7Flag co-immunoprecipitation assays to determine whether BMP5/7 or BMP6/7 heterodimers form in vivo. We tested two different commercially available BMP5 antibodies and two different BMP6 antibodies on whole embryo lysates collected at different stages and dissected organs from E13.5 embryos. We used HEK cells transfected with BMP5 or BMP6 as a positive control. While the antibodies could (in some cases) detect ectopically expressed protein, they did not detect endogenous precursor or ligand bands for BMP5 or BMP6 on immunoblots of embryo lysates or BMP7Flag immunoprecipitates. Because we cannot demonstrate that these antibodies recognize endogenous proteins, we do not want to claim that there is no interaction. We note this attempt in the Results (subsection “Biochemical analysis reveals the existence of BMP4/BMP7 heterodimers in early embryos”, last paragraph) and our inability to rule out heterodimerization between class II BMPs is described in the Discussion (Discussion, seventh paragraph. Representative test blots are shown in Author response image 1.

Reviewer #3:

[…] The current paper has several issues that need to be addressed before acceptance.1) Figure 6B should be replaced with a better figure. Why is the middle portion of the control western blot using anti-Flag antibody missing? Why are there three heavy and light chain bands in panel B? There should be only two bands. What are the additional non-specific bands marked by asterisks? Also, panels A and B need to be referred to in the figure legend.

We have replaced panel 6B with a new, and cleaner blot. The middle portion of the blot is deleted because it requires a very long exposure to detect cleaved mature BMP7Flag and the longer exposure blows out the signal for the precursor band. Thus, the two panels represent two different exposures of the same blot. We now indicate this in the Results (subsection “Biochemical analysis reveals the existence of BMP4/BMP7 heterodimers in early embryos”) and in the figure legend and have added a new supplementary figure (Figure 6—figure supplement 5) that includes the short and long exposure of the blot. The bands indicated by asterisks in the BMP4 input blot are non-specific background bands that are detected by this antibody in embryo lysates. Whereas the signal for the precursor band and ligand bands show reduced intensity in lysates from BMP4 null heterozygotes, the “non-specific” bands do not. We note this in the Results and figure legend.

2) Because the result in Figure 5 indicates that BMP7 and BMP2 form heterodimers, it would be useful to also show evidence of BMP7/BMP2 heterodimers in developing embryos by western blot, as was done in Figure 6 for BMP4/BMP7 heterodimers. Is there a reason why this cannot be easily accomplished?

Unfortunately (and several thousands of dollars later) we have been unable to identify high affinity BMP2 antibodies to successfully complete this experiment. Commercially available BMP2 antibodies do not reproducibly recognize endogenous BMP2, based on comparison of signal in lysates from wild type and BMP2 null embryos. Although we have seen what appears to be mature BMP2 immunoprecipitating with BMP7 Flag on a few occasions, this result was not reproducible in all experiments, and/or we could not detect endogenous BMP2 in lysates in the same experiment. We note this attempt in the Results (subsection “Biochemical analysis reveals the existence of BMP4/BMP7 heterodimers in early embryos”, last paragraph). Representative test blots are shown in Author response image 2.

**Author response image 2. respfig2:**